

# Isolation of soil bacteria able to degrade the anthelminthic compound albendazole

Stathis Lagos, Kalliopi Koutroutsiou and Dimitrios G. Karpouzas

Department of Biochemistry and Biotechnology, Laboratory of Plant and Environmental Biotechnology, University of Thessaly, Larissa, Viopolis, Thessaly, Greece

## ABSTRACT

Anthelmintic (AHs) veterinary drugs constitute major environmental contaminants. The use of AH-contaminated fecal material as manures in agricultural settings constitutes their main route of environmental dispersal. Once in soils, these compounds induce toxic effects to soil fauna and soil microbiota, both having a pivotal role in soil ecosystem functioning. Therefore, it is necessary to identify mitigation strategies to restrict the environmental dispersal of AHs. Bioaugmentation of AH-contaminated manures or soils with specialized microbial inocula constitutes a promising remediation strategy. In the present study, we aimed to isolate microorganisms able to actively transform the most widely used benzimidazole anthelminthic albendazole (ABZ). Enrichment cultures in minimal growth media inoculated with a soil known to exhibit rapid degradation of ABZ led to the isolation of two bacterial cultures able to actively degrade ABZ. Two oxidative products of ABZ, ABZSO and ABZSO$_2$, were detected at low amounts along its degradation. This suggested that the oxidation of ABZ is not a major transformation process in the isolated bacteria which most probably use other biotic pathways to degrade ABZ leading to the formation of products not monitored in this study. Full length sequencing of their 16S rRNA gene and phylogenetic analysis assigned both strains to the genus *Acinetobacter*. The sequences were submitted in GeneBank NCBI, database with the accession numbers OP604271 to OP604273. Further studies will employ omic tools to identify the full transformation pathway and the associated genetic network of *Acinetobacter* isolates, information that will unlock the potential use of these isolates in the bioaugmentation of contaminated manures.

## INTRODUCTION

Infections by gastrointestinal nematodes (GINs) are considered a major threat for grazing animals worldwide, leading to serious effects on their welfare and productivity (*Mavrot, Hertzberg & Torgerson, 2015*; *Kaplan, 2020*). The main strategy for prevention and treatment of GINs is the use of anthelminthic (AH) compounds (*McKellar & Jackson, 2004*; *Kaplan, 2020*). Benzimidazoles is one of the most widely used classes of synthetic AHs (*Horvat et al., 2012*). Their benzimidazole ring constitutes a very important pharmacophore moiety in drug discovery (*Zhou et al., 2016*), that has been associated with various biological activities like anticancer, antibacterial, antifungal, anti-inflammatory,

Corresponding author
Dimitrios G. Karpouzas,
dkarpouzas@uth.gr

antihistaminic, antioxidant, antihypertensive, and anticoagulant (*Tunçbilek, Kiper & Altanlar, 2009*). Benzimidazoles act as inhibitors of mitosis by binding on tubulin and thus preventing microtubule formation (*Mckellar & Scott, 1990*; *Bansal & Silakari, 2012*). Several benzimidazoles are used as AHs, like albendazole (ABZ), ricobendazole, fenbendazole, flubendazole and mebendazole (*Mckellar & Scott, 1990*).

Benzimidazole AHs are extensively metabolised in the liver of treated animals by monooxygenases belonging to the cytochrome P450 and flavin-monooxygenase families, and thus excreted through feces and urine either intact or in the form of their sulfoxides, which carry anthelminthic activity like albendazole sulfoxide (ABZSO), and sulfones (*Virkel et al., 2009*; *Miró et al., 2019*). Depending on their administration mode, 60% to 90% of the dose is excreted to urine and feces (*Halley, Jacob & Lu, 1989*; *Gottschall, Theodorides & Wang, 1990*; *Horvat et al., 2012*; *Aksit et al., 2015*). Fecal material is either left on the floor of livestock farms or stockpiled and subsequently used as manures in agricultural settings. Both these practices, combined with the proven persistence of benzimidazole AHs in feces and manures (*Prchal et al., 2016*; *Silveira Porto, Bonilha Pinheiro & Rath, 2021*) could lead to the dispersal of benzimidazole AHs in soil and their further translocation to other environmental compartments.

ABZ constitutes the most heavily used benzimidazole AHs in livestock farming. It was reported to be present in sheep feces at levels up to 12.8 and 7.7 mg kg$^{-1}$ with lower levels of ABZ transformation products, albendazole sulfoxide (ABZSO) and albendazole sulfone (ABZSO$_2$), also detected. *Lagos et al. (2021)* reported that in sheep fecal material total ABZ residues (parent compound combined with ABZSO and ABZSO$_2$) showed a DT$_{50}$ of 13 days (*Prchal et al., 2016*; *Silveira Porto, Bonilha Pinheiro & Rath, 2021*). Total ABZ residues in sheep fecal material showed a DT$_{50}$ (Degradation Time 50%, the time required for the degradation of 50% of the initial amount of the compound) of 13 days (*Lagos et al., 2021*). Once in soil ABZ is rapidly transformed to ABZSO, which also carries AH activity (*Belew et al., 2021*), and then to the AH inactive ABZSO$_2$ with DT$_{90}$ values for the total ABZ residues ranging from 41.5 to >365 days (*Lagos et al., 2022*). From soil ABZ can be either taken up by plants (*Stuchlíková Raisová et al., 2017*) and through grazing back to the animals at sublethal levels which may favor the development of drug resistance in GINs (*Navrátilová et al., 2021*) or in the form of its polar transformation products ABZSO, ABZSO$_2$ and ABZSO$_2$ amine could leach to groundwater (*Silveira Porto, Bonilha Pinheiro & Rath, 2021*). Indeed, ABZ and its transformation products constituted the most frequently detected AHs in groundwater and surface water systems in Ireland (*Mooney et al., 2021*). Considering the proven toxicity of ABZ onto non-target soil (*e.g.*, earthworm *Eisenia fetida*) (*Gao et al., 2007*) and aquatic organisms (*e.g.*, crustacean *Daphnia magna* and fish *Danio rerio*) its environmental dispersal should be mitigated.

Several treatments of fecal material like composting or anaerobic digestion have been used to reduce the load of manures to veterinary drugs. These approaches have shown variable results, so far only tested for the removal of antibiotics (*Selvam et al., 2013*; *Berendsen et al., 2018*). Recently *Turek-Szytow et al. (2020)* suggested that treatment of

manures with inorganic peroxide mixtures (PM) could effectively eliminate ABZ, although it is expected that such reactive methods could also alter the properties of manures. One interesting, promising, low-cost and non-invasive mitigation approach is bioaugmentation of fecal material or even contaminated soils with microorganisms capable of degrading ABZ and its derivatives. First attempts by *Hirth et al. (2016)* and *Hong et al. (2020)* reported interesting results on the removal of the veterinary antibiotics sulfamethazine and tetracycline from soil. In our earlier work we used a thiabendazole-degrading bacterial consortium for the bioaugmentation of feces contaminated with ABZ and other benzimidazole AHs and noted a moderate acceleration in the removal of thiabendazole, its original substrate, but a less efficient still significant removal of ABZ (*Lagos et al., 2021*). This led us to hypothesize that specialized microbial inocula tailored to the degradation of ABZ will be more efficient in the bioaugmentation of contaminated matrices. Hence, we aimed to isolate bacteria able to rapidly degrade ABZ. This was achieved through enrichment cultures from a selected soil collected from a livestock farm with regular use of ABZ which showed accelerated rates of degradation of ABZ in previous studies (*Lagos et al., 2022*, *2023*).

## MATERIALS AND METHODS

At this point we would like to address that portions of this text were previously published as part of Stathis Lagos PhD thesis (Available at: https://doi.org/10.12681/eadd/54190).

### Chemicals and growth media

An analytical standard of ABZ (98%; Tokyo Chemical Industry©, Zwijndrecht, Belgium) was used in media preparation and for analytical purposes. Analytical standard of ABZSO (98% purity) was also purchased from Tokyo Chemical Industry© (Zwijndrecht, Belgium), while $ABZSO_2$ (97% purity) was purchased from Santa Cruz Biotech© (Heidelberg, Germany). A mixture of ABZ, ABZSO and $ABZSO_2$ in acetonitrile ($1,000$ mg $L^{-1}$) were used for preparing serial dilutions ranging from $10$–$0.025$ mg $L^{-1}$ which were used to construct calibration curves for residue quantification by HPLC.

Selective mineral salts media (MSM) and its nitrogen amended version (MSMN), supplemented with ABZ as the sole C and N or the sole C source respectively, were used for the isolation of ABZ-degrading bacteria. MSM and MSMN were prepared as described before (*Karpouzas & Walker, 2000*). Growth media were spiked with a $5,000$ µg $ml^{-1}$ filter-sterilized solution of ABZ in DMSO (Molecular Biology Grade; Sigma Aldrich©, St. Louis, MI, USA) aiming to a final concentration of $5$ µg $ml^{-1}$ of ABZ in the medium. This concentration was the highest tested concentration of this compound that could be fully dissolved in the aqueous growth medium without any solubility issues and precipitation. DMSO levels in the medium never exceeded 0.1%. Growth media were also amended with 0.05% of Tween 20 to enhance ABZ solubility, as suggested in our earlier studies (*Lagos et al., 2021*). Agar plates of the aforementioned media plus ABZ and Tween 20 were prepared by addition of $15$ g $L^{-1}$ agar.

## Enrichment cultures and isolation of ABZ-degrading bacteria

To isolate ABZ-degrading bacteria, we employed enrichment cultures in MSM and MSMN supplemented with ABZ. A soil from a livestock unit in Lesvos Island, Greece, 39°16′21.4″ N 26°15′55.7″E, with history of ABZ administration and high degradation capacity towards ABZ (*Lagos et al., 2022*) was used for bacteria isolation. Prior to the onset of the enrichment cultures, the soil was repeatedly treated with ABZ (5 μg g$^{-1}$), three times on 15-day intervals to stimulate and activate the microbial community able to degrade ABZ. This concentration level is higher than the concentration levels of ABZ often detected in soils (*Thiele-Bruhn, 2003*; *Liebig et al., 2010*; *Navrátilová et al., 2023*) but it was selected based on (i) previous soil studies which have indicated that the same soil was able to degrade ABZ concentrations of 2 mg/kg or higher at an accelerated mode (*Lagos et al., 2023*) and (ii) on previous soil studies with pesticides and other organic pollutants that have indicated that such concentration levels are essential to provide the necessary carbon and energy to stimulate the specialized pollutant-degrading soil microbiota (*Topp et al., 2013*; *Rousidou et al., 2016*; *Lagos et al., 2019*). After completing the pre-treatment, 0.5 g of soil were used to inoculate triplicate bottles per medium (20 ml), while duplicated non-inoculated samples containing the same volume of each medium were used as abiotic controls. All cultures were incubated in an orbital shaker in the dark at 25 °C.

The degradation of ABZ was measured by analyzing samples at regular interval by HPLC as described below. At the point where degradation of ABZ was >70% an aliquot of each culture (0.5 ml) was transferred in fresh triplicate cultures. The same procedure was repeated for four cycles in total and at the point of 65–70% degradation of ABZ in the fourth enrichment cycle, a serial dilution was prepared and spread on MSM or MSMN agar plates amended with ABZ (5 μg ml$^{-1}$). The plates were then placed for incubation at 25 °C. After 3–4 days of incubation growing colonies were selected and transferred in the corresponding liquid media. The capacity of the selected colonies to degrade ABZ was determined at 7 days *via* HPLC. Aliquots of cultures which showed a high degradation capacity (>70% degradation in 7 days) were sub-cultured in fresh liquid media in triplicates, to confirm their degradation capacity. Only cultures exhibiting >60% degradation in 7 days were considered as positive and they were all derived from MSMN. The selected cultures were plated on MSMN + ABZ agar plates to check purity. They were then processed for DNA extraction, and further molecular analysis as described below.

## Albendazole residue analysis

ABZ was extracted from liquid media by mixing 0.5 mL of culture with 0.5 mL of acetonitrile. The mixture was vigorously vortexed for 30 sec, then filtered through 0.45-μm PTFE hydrophobic syringe filters and directly analyzed in a Shimadzu HPLC–DAD system equipped with a Grace Smart RP C18 column (150 mm × 4.6 mm) (Shimadzu Corporation, Kyoto, Japan) as described before (*Lagos et al., 2022*). Briefly, ABZ, ABZSO and ABZSO$_2$ were eluted using a gradient mobile phase of 30:70 acetonitrile:water (v/v) + 0.1% H$_3$PO$_4$ and they were detected at 227 nm. Fortification tests at three concentration levels (0.1, 1 and 10 mg L$^{-1}$) showed mean percentage recoveries for ABZ, ABZSO, ABZSO$_2$, of 91.7%, 90.4%, 95.2% (CV < 11.3%) respectively. The LOD and LOQ values for
all analytes were 0.02 and 0.05 mg L$^{-1}$ respectively. The concentrations of ABZ and its oxidation products were determined using an external calibration curve by injection of matrix-matched standard solutions (0.02–10 mg L$^{-1}$) of a mixture of all analytes. At this concentration range calibration curves showed high linearity with r$^2$ > 0.99. The repeatability of the method for all three analytes, expressed as the relative standard deviation of the recovery at all fortification levels was acceptable (≤9.4%). The reproducibility of the method for all three analytes, determined as the % recoveries obtained every 10 days for a 30-day period for aqueous matrices (minimal media) fortified at the concentration of 1 mg L$^{-1}$, was also acceptable (≤12.9%).

### Molecular identification of the ABZ-degrading bacteria

Bacteria DNA extraction was performed with the Nucleospin® Tissue kit (Marcherey–Nagel, Düren, Germany). Briefly, the near full-length (1,500 bp) 16S rRNA gene of bacterial cultures was amplified with primers 8f–1512r (*Felske et al., 1997*) as described by *Perruchon et al. (2016)*. The identity of the isolated bacteria was determined *via* cloning the PCR products, using the pGEM®-T easy plasmid vector, and sequencing of the full length 16S rRNA gene. Three clones for each isolate were Sanger sequenced and the obtained sequences were edited manually and analyzed for best match with the Basic Local Alignment Search Tool (BLAST, v.2.9.0) (*Altschup et al., 1990*). The closest relatives obtained plus an outgroup sequence were aligned with the Muscle software (*Notredame, Higgins & Heringa, 2000*). Uninformative blocks and misalignments were removed with the GBlocks software (*Talavera & Castresana, 2007*), and the sequence alignment obtained was utilized for the construction of maximum likelihood trees generated according to the general time reversible model, with gamma rate heterogeneity and accounting for invariable sites, using the PhyML software (v.3.1) (*Guindon & Gascuel, 2003*). The sequences of the clones which studied were submitted in GeneBank NCBI, database with the accession numbers OP604271 to OP604273.

## RESULTS AND DISCUSSION

### Enrichment cultures in MSM and MSMN media

The degradation of ABZ in enrichment cultures in MSM and MSMN is presented in Fig. 1. In the first enrichment cycle in both growth media degradation of ABZ was over 70% after 6 days. In both growth media and across enrichment cycles, the degradation of ABZ showed similar patterns with over 60% and 70% degradation of ABZ in MSM and MSMN respectively after 6 days from the start of each enrichment cycle (Figs. 1A and 1B). Abiotic degradation of ABZ in the non-inoculated controls in both media never exceeded 20% (Fig. 1), suggesting that the degradation of ABZ observed in the inoculated cultures is microbially driven. ABZSO (main transformation product of ABZ) and ABZSO$_2$ were detected in both inoculated and non-inoculated cultures at low levels suggesting that their formation was mostly due to abiotic degradation process, as also suggested by earlier studies (*Liou & Chen, 2018*). However, the contribution of biotic processes in the formation of ABZSO and ABZSO$_2$ cannot be entirely excluded, especially in the MSM where slightly but not significantly higher amounts of ABZSO were formed in the

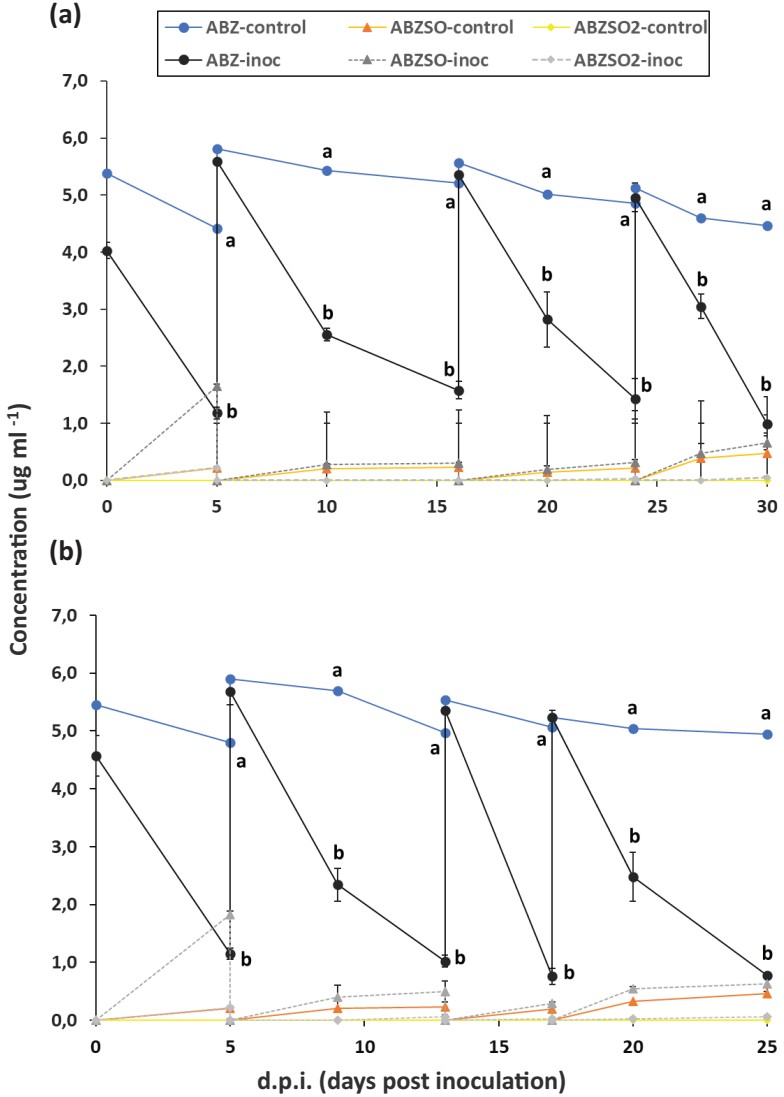

**Figure 1 Enrichment cultures with albendazole (ABZ) as a sole source of C (and/or N) in selective media MSMN and MSM.** Degradation of albendazole (ABZ), and formation and degradation patterns of its transformation products, albendazole sulfoxide (ABZSO) and albendazole sulfone (ABZSO$_2$) in four successive enrichment cycles in selective media MSM (A) and MSMN (B) either inoculated (inoc) or not inoculated (control) with a soil exhibiting enhanced biodegradation of ABZ. Each value is the mean of three replicates ± the standard deviation of the mean. At each sampling time, statistically significant differences as denoted by different lower case letters (a, b), (5% level) in the concentrations of ABZ, ABZO, ABZSO$_2$ between the control and the inoculated samples were derived by one-way analysis of variance and Tukeys *post-hoc* test.

inoculated *vs* non-inoculated cultures from the second enrichment cycle onwards. Still, it is important to note that the sum of these oxidative derivatives never exceeded 15% of the initial amount of the parent compound implying the formation of other transformation products during the biotic degradation of ABZ that were not monitored in our study. Similar studies with other xenobiotic compounds carrying thioether moieties in their molecule, like the pesticide fenamiphos, showed that soil bacteria were actively degrading
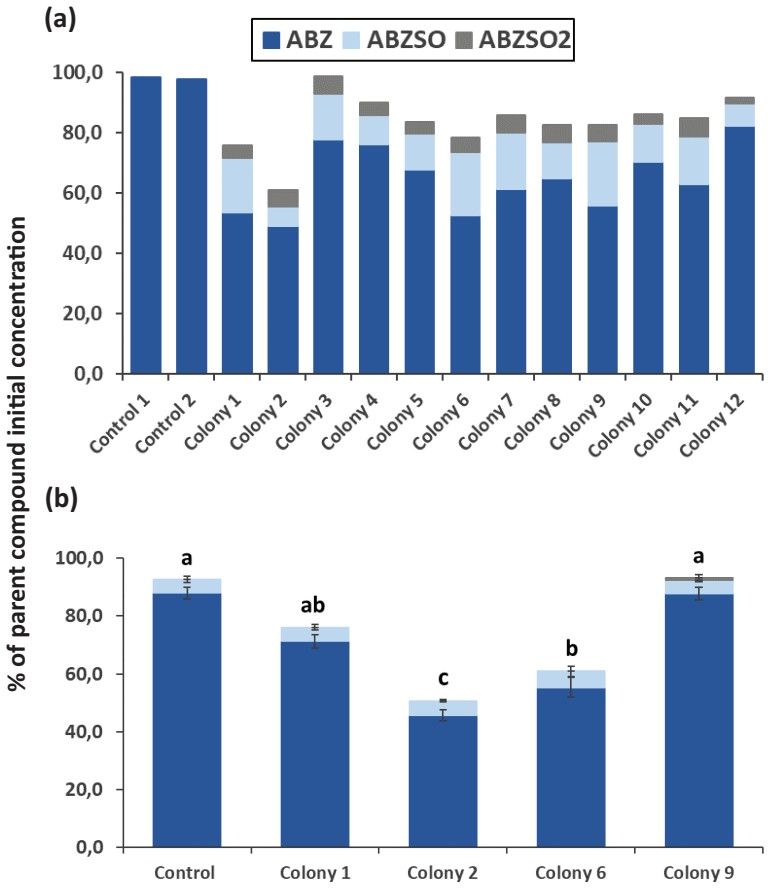

**Figure 2 The degradation levels of albendazole (ABZ) in MSM liquid cultures.** Degradation of albendazole (ABZ) and formation of its transformation products, albendazole sulfoxide (ABZSO) and albendazole sulfone (ABZSO$_2$) in MSMN liquid cultures inoculated with selected colonies and in non-inoculated controls. (A) The degradation of ABZ by colonies obtained from a first round of selection, after seven days of incubation. (B) Colonies showing promising degradation of ABZ in the first screening were tested again for their degradation capacity. Each value in the second screening round is the mean of three replicates + the standard deviation of the mean. Stacked bars designated by different letters (a, b, c) are statistically different at 5% level as determined by one way ANOVA and Tukeys *post-hoc* test.

fenamiphos and its sulfoxide and sulfone derivatives through hydrolysis rather than oxidation (*Caceres et al., 2009*; *Chanika et al., 2011*). We speculate that a similar pathway might be also active in our enrichment cultures where ABZ itself and its oxidation derivatives, ABZSO and ABZSO$_2$, are hydrolyzed through removal of the methyl carbamic moiety of the benzimidazole ring to 2-amino inactive derivatives, which were not monitored in our study and also in other relevant earlier studies. However, this speculative transformation pathway of ABZ should be further verified in follow up shotgun metabolomic analysis.

## Isolation and screening of Albendazole-degrading bacteria

After completion of enrichment cultures and plating, 20 and 12 morphologically distinct bacterial colonies were selected from MSMN + ABZ and MSM + ABZ agar plates
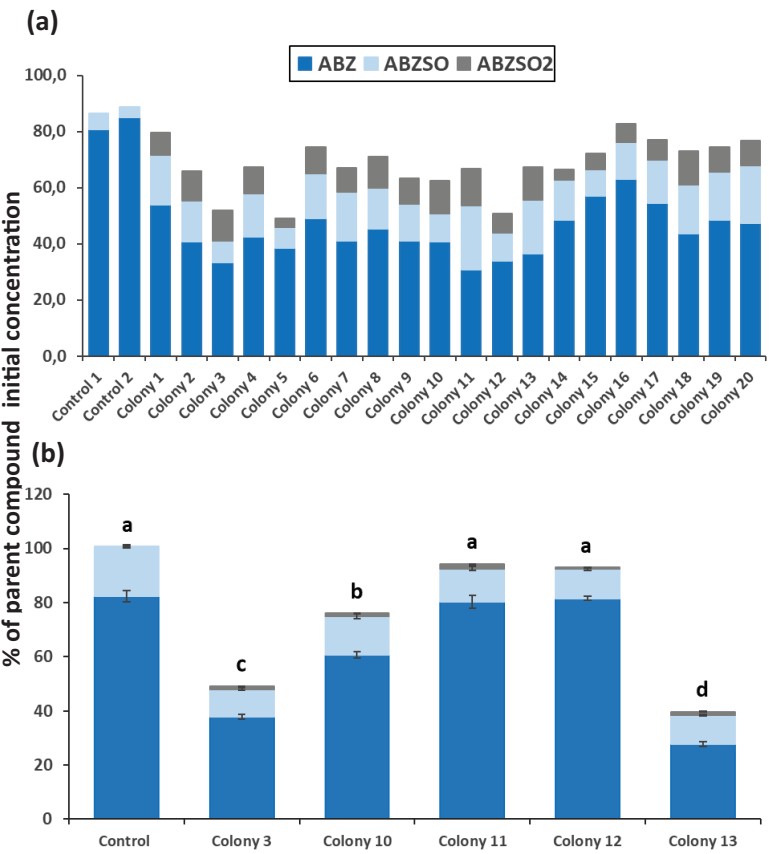

**Figure 3 The degradation levels of albendazole (ABZ) in MSMN liquid cultures.** Degradation of albendazole (ABZ) and formation of its transformation products, albendazole sulfoxide (ABZSO) and albendazole sulfone (ABZSO$_2$) in MSMN liquid cultures inoculated with selected colonies and in non-inoculated controls. (A) Degradation of ABZ by colonies obtained from a first round of selection after seven days of incubation. (B) Colonies showing promising degradation of ABZ in the first screening were tested again for their degradation capacity. Each value in the second screening round is the mean of three replicates + the standard deviation of the mean. Stacked bars designated by different letters (a, b, c) are statistically different at 5% level as determined by one way ANOVA and Tukeys *post-hoc* test.

respectively and screened for their ability to degrade ABZ (Figs. 2 and 3). Four colonies which showed more than 50% degradation, were selected from MSM cultures (Fig. 2A) but their degrading capacity was not verified in a second round of cultivation and testing (Fig. 2B). In case of MSMN, five colonies which presented >60% degradation of ABZ after 7 days of incubation were selected for further testing (Fig. 3A). From these only two cultures, named C3 and C13, maintained their high degradation capacity and exhibited >60% degradation of ABZ after 7 days of incubation (Fig. 3B). ABZ was partially transformed to ABZSO in both inoculated and non-inoculated cultures, while small amounts of ABZSO$_2$ were also detected but only in the inoculated cultures. Previous studies with fungal and bacterial isolates tested for their degrading capacity against ABZ also identified ABZSO and ABZSO$_2$ as the sole transformation products of ABZ (*Prasad, Girisham & Reddy, 2009*, *2010*). In contrast *Prasad et al. (2008)* observed, besides ABZSO
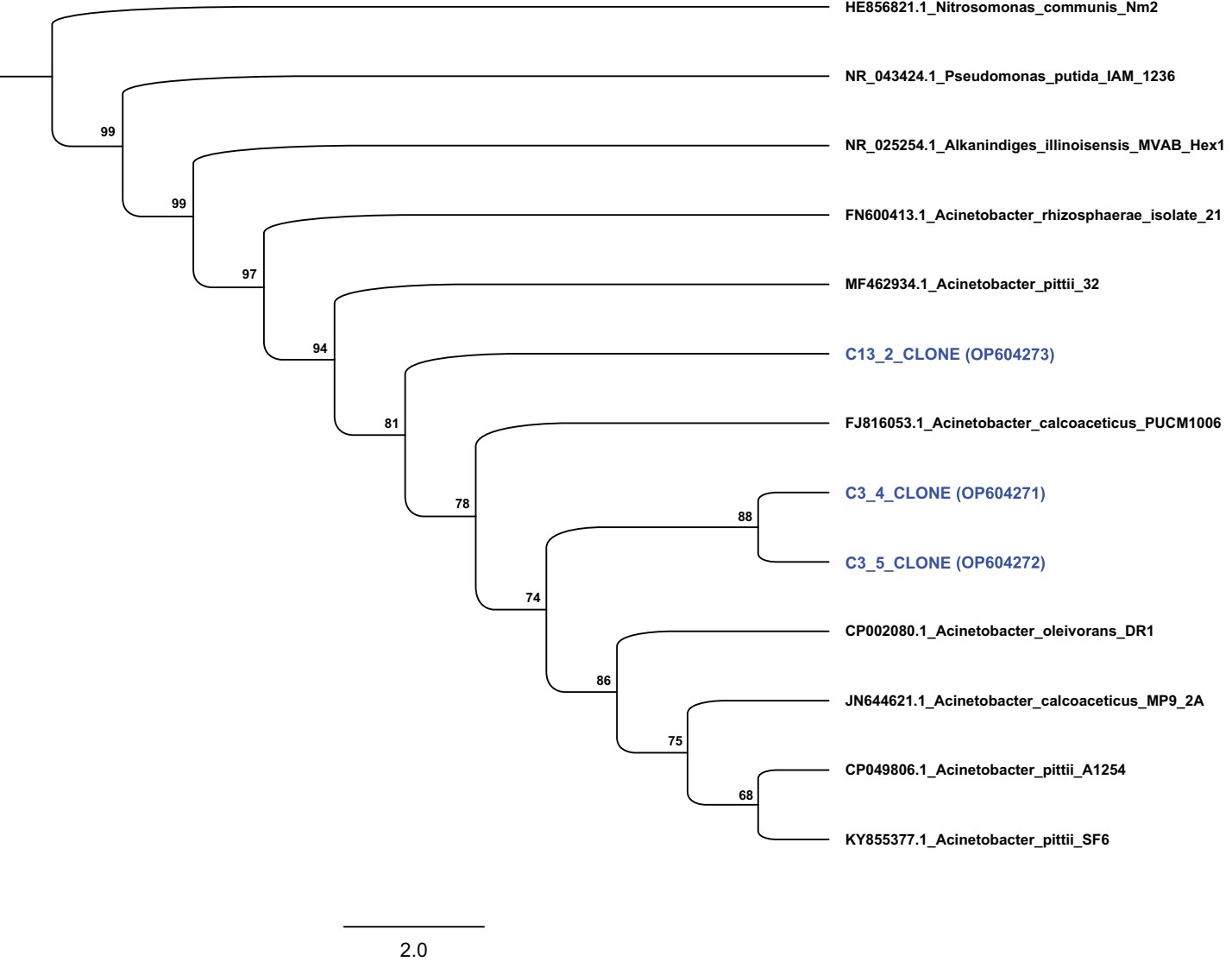

**Figure 4 Phylogenetic analysis of the sequences from selected clones based on the complete 16S rRNA gene sequence of the isolated strains degrading ABZ.** All sequences were grouped within the genus *Acinetobacter*. Thousand bootstrap replicates were run with PhyML following the GTRGAMMAI (General Time Reversible with GAMma rate heterogeneity and considering Invariable sites) model. The bootstrap support is expressed in scale from 0 to 100. The NCBI accession numbers of each clone (which presented with blue colour) are indicated.

and ABZSO$_2$, the formation of a new N-methylated derivative, produced by the degradation of ABZ by a *Cunninghamella blakesleeana* fungal strain.

## Identification of ABZ-degrading bacteria

Based on their degradation capacity against ABZ the two isolates were further identified *via* molecular means. Clone libraries, prepared from cultures C3 and C13, revealed that the phylotypes represented in these cultures showed highest sequence match to the 16S rRNA gene sequence of bacteria of the genus *Acinetobacter*. Phylogenetic analysis based on the full-length 16S rRNA gene sequence verified the assignment of the two bacterial isolates to

the genus *Acinetobacter*. Specifically, clones from culture C3 grouped with species *Acinetobacter oleivorans* and *Acinetobacter calcoaceticus*, while clones obtained from culture C13 were phylogenetically closer to *Acinetobacter pittii* (Fig. 4). However, the low bootstrap values does not allow the assignment of the two isolates to the species level. Bacteria of the genus *Acinetobacter* are ubiquitous in soil and they are characterized as metabolically versatile bacteria able to catabolize a wide range of natural compounds, implying active participation in nutrient cycling (*Jung & Park, 2015*). They are also known as efficient degraders of xenobiotic aromatic compounds like phenolic derivatives, quinones, pyridines, indoles (*Paller, Hommel & Kleber, 1995*; *Ying et al., 2007*; *Zhang et al., 2021*) and pesticides. For example, *A. calcoaceticus* and *A. oleivorans* strains were able to degrade the insecticide fipronil (*Uniyal et al., 2016*), while *Zhan et al. (2018)* and *Singh, Suri & Cameotra (2004)* isolated *Acinetobacter* stains able to degrade pyrethroids and atrazine respectively.

To date there are a few reports of microorganisms able to degrade ABZ. *Jin et al. (2013)* isolated a *Rhodococcus* strain that was able to degrade ABZ and use it as a C source, in agreement with our *Acinetobacter* isolates that degraded ABZ only in MSMN where the AH served as a sole C source. *Prasad, Girisham & Reddy (2010)* screened several bacterial strains for their capacity to oxidize ABZ to ABZSO and identified *Enterobacter aerogenes*, *Klebsiella aerogenes*, *Pseudomonas aeruginosa* and *Streptomyces griseus* stains as active degraders of ABZ. Besides bacteria, *Prasad et al. (2008)* and *Prasad, Girisham & Reddy (2009)* also isolated fungal degraders of ABZ like a *Fusarium moniliforme* strain and a *Cunninghamella blakesleeana* strain.

## CONCLUSIONS

In the present study we report the isolation of two soil bacterial isolates, identified as *Acinetobacter* spp., that were able to degrade the synthetic benzimidazole AHs ABZ. ABZSO and ABZSO$_2$ were identified as minor transformation products formed at low levels along degradation of ABZ. This suggests that the isolated bacteria use other transformation pathways, besides oxidation, to degrade ABZ. Those pathways could lead to the formation of transformation products that were not monitored in our study. Whole genome sequencing analysis and further transcriptomic or proteomic analysis will provide insights into the transformation pathway and the genetic mechanism driving the transformation of ABZ in these bacterial isolates. These information are essential before the use of these bacteria as inocula for the bioaugmentation of contaminated fecal material and soils, preventing or mitigating the dispersal of residues of benzimidazole AHs in agricultural and grassland soils.

### Funding

This work was supported by the Hellenic Foundation for Research and Innovation (HFRI) under the HFRI PhD Fellowship grant to Stathis Lagos (Fellowship Number: 530).

The funders had no role in study design, data collection and analysis, decision to publish, or preparation of the manuscript.

## Grant Disclosures
The following grant information was disclosed by the authors:
Hellenic Foundation for Research and Innovation (HFRI) under the HFRI PhD Fellowship: 530.

## Competing Interests
The authors declare that they have no competing interests.

## Author Contributions
- Stathis Lagos conceived and designed the experiments, performed the experiments, analyzed the data, prepared figures and/or tables, authored or reviewed drafts of the article, and approved the final draft.
- Kalliopi Koutroutsiou performed the experiments, prepared figures and/or tables, and approved the final draft.
- Dimitrios G. Karpouzas conceived and designed the experiments, authored or reviewed drafts of the article, and approved the final draft.

## Data Availability
The 16S ribosomal RNA gene, partial sequences, are available at GenBank: OP604271 to OP604273.

The raw measurements are available in the Supplemental Files.

The samples of the novel microbial strains are deposited in the public culture collection ATHUBA of National and Kapodistrian University of Athens: Acinetobacter sp. ATHUBA 3168 & Acinetobacter sp. ATHUBA 3169.

## Supplemental Information
Supplemental information for this article can be found online at http://dx.doi.org/10.7717/peerj.16127#supplemental-information.

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
