# Peer review of "Isolation of soil bacteria able to degrade the anthelminthic compound albendazole"

_PeerJ, doi:10.7717/peerj.16127_

## Round 0.1 · original submission · Major Revisions

"The paper did not aim to describe the new isolates but rather to report on the isolates with the specific degrading trait. To this level, and based on my view after reading the paper, the work is scientifically sound and given the scarcity of data for the fate of veterinary drugs in the environment, the paper can be considered.

Its important to address review 3 concerns about "Authors claimed “accelerated” and “slower degradation rate” but these supposed rates of degradation/disappearance of ABZ are not presented nor statistically compared… This issue must be amended, otherwise omitted."

Additional information on the HPLC conditions must be included in the revised version.

Reviewer 1 ·

Basic reporting

In this manuscript, the authors isolated soil bacteria capable of degrading the anthelminthic veterinary drug albendazole. The manuscript is well-written in English, and the background information and references are adequately introduced and cited. Overall, the manuscript is highly intriguing and the findings hold great value for the field. I only have a few specific comments.
Materials and Methods
1. L120-122: the Atuhors pretreated the soil with ABZ to stimulate and activate the microbial community able to degrade ABZ.
I am wondering that if the soil was collected from ABZ polluted farms, bacteria which can degrade ABZ should be already there, why need to pretreat? For enriching the target bacteria? And also increased the bacteria mutation with ABZ pretreatment. Do authors test the non-pretreat soil for bacteria isolation? If have this kind of data please also add them as a control group.
Results:
3. All figures need to be optimized with a better dpi.
4. There is no statistical analysis for the collected data. Please do a statistical analysis for your data.
5. L206: If the authors can show the MIC of isolated ABZ-degration bacteria for ABZ and the highest concentration of ABZ of isolated Acinetobacter can degrade would be great for this manuscript.

Experimental design

Listed as 1. basic report.

Validity of the findings

Listed as 1. basic report.

Additional comments

Listed as 1 basic report.

Reviewer 2 ·

Basic reporting

The manuscript is well experimented with supporting data . I suggested to remove "anthelminthic veterinary drug " from the title.

Experimental design

Well designed and expermented.

Validity of the findings

Very good

Additional comments

Title should be modified.
Abstract and conclusion should be rewritten for more clarity and further importance of this work.

Reviewer 3 ·

Basic reporting

This manuscript reports on the isolation of soil microorganisms able to degrade the anthelmintic drug albendazole. The topic of the research is of environmental concern because anthelmintics and other drugs used in livestock are eliminated through urine or faeces to the environment and could contaminate soil, surface/ground water, and could also be absorbed by plants, including crops. The manuscript is well written, and the methods are sound. The main concern is on the observed degradative fate of the anthelmintic, which appear to be “converted” in other products rather than in those metabolites that are well known to be formed in animals and man after being treated with the drug.

Experimental design

no comment

Validity of the findings

no comment

Additional comments

INTRODUCTION

Line 49: Add Lubega, G.; Prichard, R. (1991). Interaction of benzimidazole anthelmintics with Haemonchus contortus tubulin: binding affinity and anthelmintic efficacy. Experimental Parasitology, 73, 203-209. Is a more appropriate reference.

Lines 52-54: Generally, BZDs anthelmintics are intensively, rather than partially, metabolized by animals and human beings. For instance, the pre-systemic metabolism of albendazole and triclabendazole (an halogenated BZD) in the liver determines that the parent drugs are not detected in the systemic circulation of treated animals. Please see the following references that may improve the introduction (Inhibition of cytochrome P450 activity enhances the systemic availability of triclabendazole metabolites in sheep. Virkel G, et al. J Vet Pharmacol Ther. 2009 Feb;32(1):79-86. doi: 10.1111/j.1365-2885.2008.01006.x; Comparative hepatic and extrahepatic enantioselective sulfoxidation of albendazole and fenbendazole in sheep and cattle. Virkel G, et al. Drug Metab Dispos. 2004 May;32(5):536-44. doi: 10.1124/dmd.32.5.536; In vitro inhibition of the hepatic S-oxygenation of the anthelmintic albendazole by the natural monoterpene thymol in sheep. Miró V, et al. Xenobiotica. 2020 Apr;50(4):408-414. doi: 10.1080/00498254.2019.1644390.

Lines 60-63: Please add a reference on the levels of ABZ and metabolites in faeces.

Lines 63-63: Please, re-write following this suggestion… Total ABZ residues (parent compound combined with ABZSO and ABZSO2) in sheep faecal material showed a DT50 of 13 days (Lagos et al., 2021)
Line 64: DT50? Please define.

Line 69: “…which may favour…” Please, add the word “may” because, in the best of the scientific knowledge at the present, the generation of parasitic resistance as consequence of environmental levels of anthelmintic drugs has not been demonstrated.

MATERIALS AND METHODS

Lines 111 and 121: Please, explain the rationale for the selection of 5 µg/mL or g ABZ concentration in the culture medium. Is there any evidence that this concentration stimulates and activates microbes able to degrade ABZ. Please, include this kind of information in the manuscript.

Lines 140-148: This section must be improved. Add information on the chromatographic conditions and validation (in addition to recoveries, add information on linearity, lack of fit tests, reproducibility, and repeatability).

RESULTS AND DISCUSSION

As a general comment, the degradation of ABZ did not correlate with the percentages of the metabolites formed (e.g., Figure 1). For instance, at days 5, 10, 15, 24 and 30 the % of ABZ remaining are between 20-40% whilst the % of the metabolites formed are below 20% and should be around 80-60%. Is there any reason for this observation? Could ABZ be degraded into other metabolites? Could ABZ be adsorbed to the glassware (i.e., incubation bottle)? All these possibilities must be tested, analysed, and discussed more deeply. Authors also measured the production of the metabolites formed in animals but the degradation of ABZ observed in the current research seems to be consequence of its conversion into other degradative products which, in addition, could also be of environmental concern.
Figure 1: Data must be presented in ug/mL (nmol/mL should be better) rather than in percentages.

Lines 174-175: Authors claimed “accelerated” and “slower degradation rate” but these supposed rates of degradation / disappearance of ABZ are not presented nor statistically compared… This issue must be amended, otherwise omitted.

---

## Round 0.2 · Minor Revisions

Dear Authors

Could you please address the concern raised by reviewer 3 about "degradative fate of the anthelmintic was not completely clarified nor accordingly discussed in the revised version."

Reviewer 1 ·

Basic reporting

The authors have addressed my questions well. I have no more comments.

Experimental design

The authors have addressed my questions well. I have no more comments.

Validity of the findings

The authors have addressed my questions well. I have no more comments.

Reviewer 3 ·

Basic reporting

Authors have addressed most of the comments and suggestions. However, the main concern on the observed degradative fate of the anthelmintic was not completely clarified nor accordingly discussed in the revised version. There is an overestimation of the conversion of the anthelmintic into its sulfo-metabolites while the main degradative pathway was not studied nor properly speculated in the discussion.

Experimental design

No comments

Validity of the findings

Apart of the isolation of bacteria degrading the anthelmintic ABZ, The main outcome of the current work is that the degradation of ABZ in soil is not a complete consequence of it conversion into sulfo-metabolites that are produced in treated animals and could also be found in the environment.

Additional comments

Line 58 (suggested sentence): Benzimidazole anthelmintics are extensively metabolised in the liver of treated animals by monooxygenases belonging to the cytochrome P450 and flavin-monooxygenase families.

Lines 110-114: In my view it is not necessary this statement that could be cited as a footnote in the first page as follows: “Part of this work was previously published in the PhD Thesis by Stathis Lagos (“The role of soil microogranisms in the mitigation of the environmental deterioration imposed by the use and dispersal of anthelminthic veterinary drugs”, 2023. National Archive of PhD Theses. Available at: https://doi.org/ 10.12681/eadd/54190). Although authors explained the rationale for the concentrations of ABZ employed in the rebuttal note, this issue has not been included in the manuscript. Please, give a clear rationale for the concentrations of the anthelmintic assayed and explain if those concentrations used are within the environmental concentrations found (i.e., those concentrations previously measured in soil).

Line 222: “The levels of ABZSO and ABZSO2 formed were slightly higher in the inoculated compared to the non-inoculated cultures…” From the results showed in Figure 1, the “degradation” of ABZ in soil seems to be higher in inoculated vs. control. However, it is not clear why authors stated that the levels of the sulfo-metabolites are “slightly-higher”; this appear to occur only at day 5, particularly for ABZSO. In my view, authors are overestimating the degradation of ABZ into the sulfo-metabolites while, as is evident in view of the obtained results, that the main route of bacterial degradation is other. This issue must be discussed accordingly in the next revised version.

---

## Round 0.3 · accepted · Accept

I can confirm that all reviewers' comments have been addressed.